# Development and validation of a Japanese translation of the K-SF-42

Tetsuya Kawamoto[1,2]*, Satoru Kiire[3], Rachel Zambrano[4], Mateo Peñaherrera-Aguirre[4], Aurelio José Figueredo[4]

1 Faculty of Letters, Kokushikan University, Tokyo, Japan, 2 Graduate School of Education, The University of Tokyo, Tokyo, Japan, 3 Faculty of Law, Osaka University of Economics and Law, Osaka, Japan, 4 Department of Psychology, University of Arizona, Tucson, AZ, United States of America

* ktetsuya@kokushikan.ac.jp

**Data Availability Statement:** The present data and R code were openly available at <https://osf.io/3nmxv/>.

## Abstract

In this study, we conducted the translation and validation of the K-SF-42 in Japan (Figueredo, 2017). The K-SF-42 is a new short form of the Arizona Life History Battery. We obtained empirical evidence that the original seven-factor structure could be applied to the Japanese translated version of K-SF-42 (K-SF-42-J). We also observed good internal consistency of the seven scales of K-SF-42-J. The multi-group confirmatory factor analysis findings suggest that the K-SF-42-J can be used in both sexes and diverse age groups. The K-SF-42-J scales showed similar correlates as the English original, with higher scores of other life history strategy measurement, trait emotional intelligence, well-being, and cultural and social resources in childhood. Use of the K-SF-42-J will allow researchers with Japanese speaking samples to integrate their findings with the existing life history strategy research literature. The brevity of the K-SF-42-J will be appealing to researchers who are concerned about taxing the time and motivation of their participants.

## Introduction

In recent years, Life History Theory [1, 2] has produced novel and testable hypotheses in psychological research. Life history theory postulates that organisms optimally allocate their total bioenergetic and material resources between *somatic effort* and *reproductive effort*. The former is characterized by devoting resources to the continued survival and maintenance of the individual organism, and the latter is characterized by devoting resources to the reproduction and support of offspring. Furthermore, *reproductive effort* is divided into two components: *mating effort* and *parental effort*. *Mating effort* reflects resources consumed in obtaining and retaining sexual partners. *Parental effort* refers to using resources to enhance the survival of offspring.

Life history theory was originally adopted by evolutionary biologists to explain phenotypic and behavioral differences *between* species. The contrast between *r*- and *K*-selected species [3] illustrates the variability of species-specific energy allocation based on life history theory. However, researchers have recently adopted life history theory to explain individual differences *within* species. The contrast between *r*-selected (also called *fast* life history) and *K*-selected (also called *slow* life history) strategies is evident in individual life history strategies. The

**Funding:** This research was supported by JSPS KAKENHI Grant Number JP17K13921. The funders had no role in study design, data collection and analysis, decision to publish, or preparation of the manuscript.

**Competing interests:** The authors have declared that no competing interests exist.

specific allocation of resources used by an individual (*r*- or *K*- selected) is considered adaptive given the local environmental conditions associated with a particular individual or group [4]. The specific factors in a local environment (harsh or safe; unpredictable or predictable; and uncontrollable or controllable) have influences on adaption of life history strategies [5]. Individuals who are raised in a harsh, unpredictable, or uncontrollable environment tend to develop *faster* life history strategies. They will distribute their resources more toward *mating effort* than to the *parental effort*, and more toward *reproductive effort* than to the *somatic effort*. However, individuals who are raised in a safer, more predictable, or more controllable environment are likely to develop *slower* life history strategies. Where they allocate their resources more toward *somatic effort* than to the *reproductive effort*, and more toward *parental effort* than to the *mating effort*.

This concise theory produces many novel predictions about human behaviors throughout the lifetime, which have been empirically tested in multiple scientific disciplines [6]. Previous research has shown that life history strategies are associated with general health, personality, cognitive ability, social relationships, demographic variables, and so on [7–11]. As mentioned previously, individual differences in life history strategies are partly determined by specific factors of an early environment [5]. Based on such evidence, one can conclude that, life history theory is an encompassing integrative framework for understanding lifelong human development.

To make progress in research on human life history strategies, one must first establish a psychometrically valid measurement tool. Previous research has offered well-validated scales measuring human life history strategy. Figueredo developed a questionnaire, the Arizona Life History Battery (ALHB), which is designed to measure life history strategy in humans [12]. The ALHB derived from cognitive and behavioral scales of the National Survey of Midlife Development in the United States (MIDUS) [13]. The ALHB is a 199-item psychometric questionnaire measuring human life history strategy. It contains seven broad life history domains: (a) *insight, planning, and control*; (b) *mother/father relationship quality*; (c) *family social contact and support*; (d) *friends social contact and support*; (e) *romantic partner attachment*; (f) *general altruism*; and (g) *religiosity*. These seven major domains could largely cover the areas of psychosocial resource investment in life history theory [14]. Based on the life history theory, a *slower* life history strategy reflects higher levels of these seven domains, and a *faster* life history strategy reflects lower levels of these seven domains.

## The Mini-K and its characteristics

In addition to the seven major life history domains, the ALHB includes a short domain-general form that is designed to measure the overall life history strategy general factor directly. The short-form measure, entitled the Mini-K, is composed of two to three items on each of the seven domains associated with human life history strategy. Due to the Mini-K including only 20 items, it can be used to measure a general higher-order life history strategy without participants' response burden [14]. A recent meta-analytic study has shown convergent correlations between the Mini-K score and other related measures, including personality, cognitive ability, psychosomatic health, and social relationships, which supports its adequate construct validity [6]. In addition, the Mini-K has been utilized in different international samples, which shows that the Mini-K is a cross-culturally well-validated measurement [15–21].

However, the Mini-K has two methodological shortcomings [22]. First, the internal consistency of the Mini-K was not adequate due to the relatively low number of its items (*n* = 20). This may also be due to the wide breadth of life history strategy construct that is being summarized by the single-scale Mini-K. Human life history strategy includes diverse components

including cognitive ability, family relationships, altruistic behaviors, and so on. As the Mini-K assesses all seven factors associated with life history theory by using 20 items, the item contents can inevitably become heterogeneous in content and internally inconsistent. Second, we couldn't obtain the subscale levels of the ALHB only using the Mini-K. As the Mini-K was designed to measure *global* life history strategy in a single dimension, it only includes a couple of items per each life history strategy domain. Figueredo et al. noted that it is not recommended to use the Mini-K when life history strategy is the main focus of the research [22].

## The K-SF-42

According to Figueredo et al., the long-form ALHB is preferred to use in research which mainly focuses on life history strategy, as opposed to the short forms. However, the ALHB includes 199 items, which imposes a large response burden on the participants. Thus, a new short form of the ALHB was developed [22]. The K-SF-42, which consists of 42 items, derives from the ALHB. In the process of developing the K-SF-42, six items were selected from each of the seven domains of the ALHB. Previous research has shown its good reliability and validity, and its usefulness in psychological research [23–26]. By design, a high degree of cross-cultural validity was assured by sampling only the items that performed comparably well in different samples from Australia, Italy, Mexico, Singapore, and the USA, by selecting those with the highest cross-sample geometric mean part-whole item correlations from each subscale [22].

Compared to the Mini-K, the K-SF-42 is a much better measurement. The structure of the K-SF-42 allows for researchers to obtain a subscale score for each of the seven subscales. The calculated seven scores maintain the breadth of life history strategy constructs [22]. To examine the detailed associations between life history strategy and other variables, calculating the seven life history strategy scores is very useful.

Up to now, the Mini-K has been applied to cross-culturally diverse populations, which adequately supports its cross-cultural validity. However, the K-SF-42 has not been applied to samples as diverse as the Mini-K, since it is relatively new short-form scale. Specifically, the K-SF-42 has been used in Asia, but only in Singapore [22]. To extend previous research on life history strategy, it is important to translate the K-SF-42 into other languages and test it on culturally diverse populations. Previously, a validated version of the K-SF-42 in Japanese was not available. Thus, the goal of this study is to translate the K-SF-42 into Japanese and to validate the translated version, for further use in future cross-cultural investigations.

## Present study

To this end, we translated the K-SF-42 into Japanese using the back-translation method [27]. Then we analyzed the factor structure of the Japanese version of K-SF-42 (K-SF-42-J). Considering the previous research findings [22], we expected that a seven-factor confirmatory factor analysis (CFA) model show a good fit to the data. In addition, because it is necessary to investigate the reliability and validity of the K-SF-42-J, we aimed to investigate its scale reliability in terms of internal consistency and its construct validity via nomological associations with related variables. In the K-SF-42-J items, the six items of *romantic partner attachment* are reverse-coded. The present study reverse-scored those six items to estimate secure adult romantic attachment. After doing this, individuals with *slower* life history strategy are expected to score higher in all seven life history strategy domains (*insight*, *planning*, *and control*; *mother/father relationship quality*; *family social contact and support*; *friends social contact and support*; *romantic partner attachment*; *general altruism*; and *religiosity*). The Mini-K score reflects a general higher-order life history strategy, it should be positively correlated with all the scores of K-SF-42-J (**Hypothesis 1–1**). Additionally, Figueredo et al. have shown that K-

factor, which is extracted from the ALHB subscales, is highly positively correlated with general factors of personality and of covitality [8]. Based on this finding, we, therefore, expect that emotional intelligence, which is roughly equal to general factor of personality [28], and psychological well-being to be positively correlated with all the factor scores of K-SF-42-J **(Hypothesis 1–2)**. Moreover, development of life history strategy is affected by environment in early childhood [5, 29]. Hence, we can predict that the abundance of cultural and social resources in childhood is positively associated with all the factor scores of K-SF-42-J **(Hypothesis 1–3)**.

In addition to these, we can predict that the subscales of the K-SF-42-J are associated with the following variables: effortful control; habitual exercise; procrastination; parenting styles of caregivers in childhood; social support; attachment style to their romantic partners; long-term mating orientation; unrestricted sociosexual orientation; altruistic behaviors; and spirituality. Effortful control reflects the core component of self-regulation and ability to inhibit prevailing behavior, as well as regulate emotional, behavioral, and physiological responses during this process [30]. Since individuals with *slower* life history strategy are characterized in part by the function of long-term survival and fostering long-term cooperative alliances, effortful control reflects the ability to inhibit prevailing behavior, to perform something that has to be done, and to regulate interrupting responses for the future is substantial for them. Specifically, among the seven subscales of the K-SF-42-J, we could predict the significant positive association between *insight*, *planning*, *and control* and effortful control **(Hypothesis 2–1)**.

Exercise denotes physical activity that is purposefully done to maintain or improve physical fitness [31]. As individuals with slower life history strategies tend to allocate their bioenergetic resources toward *somatic effort*, in conjunction with the previous research that has shown habitually engaging in exercise is associated with higher level of cognitive ability [32], we could expect that the significant positive association between *insight*, *planning*, *and control* and habitual exercise **(Hypothesis 2–2)**.

Procrastination is defined as the failure of self-control with intentional delay in some tasks that should be solved [33]. Procrastination has been negatively related to impulse control and pursuit of long-term goals. Previous research has shown that procrastination is part of a *fast* life history strategy [34]. Hence, we can hypothesize that procrastination is negatively associated with *insight*, *planning*, *and control* in the K-SF-42-J **(Hypothesis 2–3)**.

Parental styles of caregivers affect their childrearing behaviors, which has an influence on the development of life history strategies [29]. Warm, affectionate, responsive, and supportive parenting style is linked to *slower* life history strategy of an individual's children, while, on the other hand, cold, affectionless, rejecting, and inconsistent parenting leads to *faster* life history strategy in an individual's children. Thus, we could predict that consistently sensitive attentive parenting style is positively associated with *mother/father relationship quality*, and that inconsistent and insensitive parenting style is negatively associated with *mother/father relationship quality* **(Hypothesis 2–4)**.

Social support is defined as an instrumental and expressive assistance derived from interpersonal relationships. Perceived social support could be provided by three specific sources—the family, friends, and significant others [35]. More perceived social support implies an individual derives from a supportive environment, which is related to *slower* life history strategy. With this in mind, we expect that social support from family members is positively associated with *family social contact and support* in the K-SF-42-J, and that social support from friends and significant others is positively associated with *friends social contact and support* in the K-SF-42-J **(Hypothesis 2–5)**.

Adult attachment reflects the emotional bonds between romantic partners [36, 37]. Anxious attachment refers to a fear of being abandoned by one's romantic partner, and avoidant

attachment denotes reluctance to be close to the significant person. Individuals with secure attachment show low levels of attachment anxiety and avoidance. Those with insecure attachment score high in either attachment anxiety and avoidance, or both. Research has shown that secure attachment is linked to *slow* life history strategy, and that insecure attachment is linked to *fast* life history strategy [38]. In addition, adult romantic attachment somewhat reflects alternative reproductive strategies rather than attachment processes [39]. Specifically, a high level of avoidant attachment is positively linked to short-term mating orientation or unrestricted sociosexuality [40], and negatively lined to long-term mating orientation [41]. Based on these previous findings, attachment anxiety, avoidance, and unrestricted sociosexual orientation are expected to be negatively associated with *romantic partner attachment* (note that a higher score reflects more secure attachment after reverse-scoring), and long-term mating orientation is expected to be positively associated with *romantic partner attachment* (**Hypothesis 2–6**).

Altruistic behavior is defined as behavior that provides benefits to others rather than the self, often at the sacrifice of one's own resources [42]. Previous research has shown that individuals with *slower* life history strategy tend to show more altruistic or prosocial behaviors [14], based on this we can predict that altruistic behavior is positively associated with *general altruism* in the K-SF-42-J (**Hypothesis 2–7**).

Spirituality has been defined as the search for the meaning and purpose in life and the existence of the sacred [43]. Additionally, spirituality is dynamically interconnected with religiosity within the larger context of personal, social, and situational forces. Individuals with *slower* life history strategy are likely to show higher levels both in spirituality and religiosity [14]. Thus, spirituality is expected to positively correlate with *religiosity* in the K-SF-42-J (**Hypothesis 2–8**).

To test these hypotheses, we conducted a cross-sectional survey with six different samples of Japanese adults.

## Materials and methods

### Ethical consideration

The present survey (approval number 18–355) was approved by the institutional review boards of the University of Tokyo, to which the first author belonged at the time of conducting this survey. Since only the participants who agreed to participate in the present survey answered the questionnaire anonymously, no written or verbal consent was taken.

### Participants and procedures

Due to the large number of measurement items in this study, we divided the scales utilized in this study into six questionnaires, each containing different scales except for the K-SF-42-J. In addition, six different samples of approximately 220–320 individuals were independently collected in order to apply a different questionnaire to each sample. The total number of participants were 1440 Japanese adults ($n$ = 781 for men; $n$ = 659 for women) who live in Japan. The mean age of all participants was 43.61 ($SD$ = 10.03, range = 20–59). The detailed number of participants in each sample is shown in Table 1. All participants were members of an online research panel service provided by GMO research (https://gmo-research.com/?_fsi= 6bWRouVa), a major marketing research company in Japan. This research panel includes approximately two million registered people, from which we collected the present sample.

The six different questionnaires were web-based and commonly included the K-SF-42-J. Although the other scales were different each other, the total number of items in each questionnaire was approximately one hundred. The detailed composition of each questionnaire

**Table 1. Sample characteristics.**

| Sample | Men | Women | K-SF-42-J | Mini-K-J | TEIQue-SF | CSR | PBI | STS | GPS | HE | ECSA | WHO5 | LTMO | SOI-R | ECR-RS | SRAS-DR | MSPSS |
|---|---|---|---|---|---|---|---|---|---|---|---|---|---|---|---|---|---|
| A | 123 | 100 | X | X | X | | | | | | | | | | | | |
| B | 173 | 157 | X | | | X | X | | | | | | | | | | |
| C | 120 | 103 | X | | | | | X | X | X | | | | | | | |
| D | 130 | 93 | X | | | | | | | | X | X | | | | | |
| E | 117 | 104 | X | | | | | | | | | | X | X | X | | |
| F | 118 | 102 | X | | | | | | | | | | | | | X | X |

*Note*. X denotes that the scale was measured in that sample. K-SF-42-J = the Japanese translated version of K-SF-42; Mini-K-J = the Japanese translated version of Mini-K; TEIQue-SF = Trait Emotional Intelligence Questionnaire-Short Form; CSR = Cultural and social resource items; PBI = Parental Bonding Instrument; STS = Self-Transcendence Scale; GPS = General Procrastination Scale; HE = habitual exercise; ECSA = Effortful Control Scale for Adults; WHO5 = WHO5 Well-being Index; LTMO = long-term mating orientation scale; SOI-R = Revised Sociosexual Orientation Inventory; ECR-RS = Experience in Close Relationship-Relationship Structure; SRAS-DR = Self-Report Altruism Scale Distinguished by the Recipient; and MSPSS = Multidimensional Scale of Perceived Social Support.

was presented in Table 1. Participants received an invitation for the present study via email on July 4, 2019. This invitation contained information about informed consent, reward for participation, and a hyperlink to the web-based survey. To prevent each participant from answering more than one questionnaire form, each participant randomly received only one hyperlink of a questionnaire form. The participants read the instructions on the top page of the survey form regarding the informed consent and the voluntary nature of survey participation. Then, only those who agreed to participate in the survey clicked on the link to proceed to the survey content page. Consent to participate in the survey was confirmed by proceeding past the top page of the form. The participants who completed the questionnaire received approximately $0.5 USD as a reward for participation. The present survey (approval number 18–355) was approved by the institutional review boards of the University of Tokyo, to which the first author belonged at the time of conducting this survey.

## Measures

**Life history strategy.** The present study used a Japanese translation of K-SF-42 developed by Figueredo et al. to measure life history strategy [22]. The procedure of translating the K-SF-42 was as follows. First, the first and second authors of the present paper independently translated the K-SF-42 into Japanese and reconciled the translated sentences. Second, after the first and second author completed the translation, the K-SF-42-J was retranslated to English by translators and native English speakers, who are familiar with psychology, at a leading language service company (Crimson Interactive; http://www.crimsonjapan.co.jp/). Third, after receiving the reverse translation from the company, the first and last authors reconciled the differences between the original and reverse translated English sentences. Lastly, after some discussion regarding the translation, the K-SF-42-J was finalized. As with the original, the K-SF-42-J consisted of 42 items. The K-SF-42-J measures seven components of life history strategy—*insight, planning, and control*; *mother/father relationship quality*; *family social contact and support*; *friends social contact and support*; *romantic partner attachment*; *general altruism*; and *religiosity*—with six items for each component. The participants were asked to answer the four scales of *insight, planning, and control*; *romantic partner attachment*; *general altruism*; and *religiosity* on a seven-point Likert scale from disagree strongly (−3) to agree strongly (3). The remaining three scales of *mother/father relationship quality*; *family social contact and support*; and *friends social contact and support* were rated on a four-point Likert scale from not at all (0) to a lot (3). The six items of *romantic partner attachment* remained reverse-coded. After doing

this, higher scores reflect *slower* life history strategy on all scales in the K-SF-42-J. The K-SF-42-J was applied to all participants (Samples A–F).

Life history strategy was also measured using the Japanese version of the Mini-K (Mini-K-J) [8, 18] in the Sample A. The Mini-K-J consists of 20 items and the participants in the Sample A answered each one using a seven-point Likert scale ranging from disagree strongly (−3) to agree strongly (3). A higher score refers to a *slower* life history strategy. Internal consistency of the Mini-K-J was adequate (Cronbach's α = 0.87, 95%CI = [0.86, 0.88]).

**Trait emotional intelligence.** Trait emotional intelligence was measured with the Japanese version of Trait Emotional Intelligence Questionnaire-Short Form (TEIQue-SF) [44, 45] in the Sample A. The TEIQue-SF consists of 30 items that measures global trait emotional intelligence (EI), though it can also be used to measure the four subscales of trait EI (Well-Being, Self-Control, Emotionality, and Sociability). Since we aimed to investigate the associations between the subscales of the K-SF-42-J and global trait EI in the present study, we only calculated the score of global trait EI. Participants responded to the items using a seven-point Likert scale ranging from (1) completely disagree to (7) completely agree. A higher score refers to a higher level of trait EI. Internal consistency of the TEIQue-SF was adequate (α = 0.92, 95% CI = [0.91, 0.92]). The sample items of the TEIQue-SF are as follows: "Expressing my emotions with words is not a problem for me," "On the whole, I'm pleased with my life," "On the whole, I'm able to deal with stress," and "I can deal effectively with people."

**Psychological well-being.** Psychological well-being was measured with the WHO5 Well-being Index [46] in Sample D. This study used the authorized Japanese translations of the WHO5 (https://www.psykiatri-regionh.dk/who-5/Pages/default.aspx). To explore mood and daily behaviors during the previous two weeks, participants were asked to rate on a six-point Likert scale ranging from (0) at no time to (5) all of the time. A higher score reflects a higher level of psychological well-being. Internal consistency of the WHO5 was adequate (α = 0.91, 95%CI = [0.90 0.92]). The sample items of the WHO5 are as follows: "I have felt cheerful and in good spirits" and "I have felt active and vigorous."

**Cultural and social resources in childhood.** Cultural and social resources (CSR) in childhood were measured with 10 items collected from previous research in Japan [47] in Sample B. Six of the ten items measured the richness of cultural activities and cultural assets, and the remaining four items measured the richness of social relationships in childhood. The cultural resource items included the following: "My parents often read books to me," "My parents often read books or newspapers," "My parents made homemade sweets at home," "My parents took me to a museum," "My parents often listened to classical music," and "There were a lot of books (excluding magazines and comics) in the house." The social resource items included the following: "I talked with my family about what happened at daycare, kindergarten, or elementary school," "It was fun to meet my friends at daycare, kindergarten, or elementary school," "I usually eat dinner with my family on a weekday," and "I often took part in local events and festivals." Participants were asked to answer the CSR items as follows: "*Please remember your childhood (nursery school, kindergarten, and elementary school). The following items ask about your childhood. Please select the applicable options.*" The CSR items were answered on a four-point Likert scale ranging from (0) *not true at all* to (3) *very true*. The cultural resource and social resource scores were calculated by summing the six-item and four-item responses, respectively. High scores reflect that they had more cultural and social resources during their childhood. Internal reliabilities of the CSR items were acceptable (α = 0.75, 95%CI = [0.73, 0.77] for cultural resources; α = 0.78, 95%CI = [0.76, 0.79] for social resources).

**Effortful control.** Effortful control was measured with the Japanese version of Effortful Control Scale for Adults (ECSA) [48, 49] in Sample D. The ECSA consists of 35 items measuring global effortful control, though it can also be used to measure the three subscales:

inhibitory control, activation control, and attentional control. As we aimed to investigate the associations between *insight, planning, and control* of the K-SF-42-J and global effortful control in this study, we only calculated the score of global effortful control. Participants answered the items using a four-point Likert scale ranging from (1) *not very true of me* to (4) *very true of me*. A higher score refers to a higher level of effortful control. Internal consistency of the ECSA was adequate (α = 0.89, 95%CI = [0.88, 0.90]). The sample items of the ECSA are as follows: "When I am trying to focus my attention, I am easily distracted," "It is easy for me to hold back my laughter in a situation when laughter wouldn't be appropriate," "As soon as I have decided upon a difficult plan of action, I begin to carry it out," and "When interrupted or distracted, I usually can easily shift my attention back to whatever I was doing before."

**Habitual exercise.** Habitual engagement in exercise was measured with the Japanese version of the Moderate-to-Vigorous Physical Activity Habit Strength Scale (MVPA-HSS) [50, 51] in Sample C. This scale consists of 17 items that are rated on a six-point Likert scale ranging from (1) *not true for me* to (6) *very true for me*. A higher score reflects a higher level of habitual engagement in exercise. Internal consistency of this scale was adequate (α = 0.97, 95% CI = [0.97, 0.97]). The sample items of the MVPA-HSS are as follows: "If I don't exercise I feel restless," "I exercise for the same amount of time in each session," "Seeing other people exercise motivates me to be more active," and "I exercise without having to think about it."

**Procrastination.** Procrastination was measured using the Japanese version of General Procrastination Scale (GPS) [52, 53] in Sample C. Though the original GPS consists of 20 items, Hayashi reduced them to 13 items in the development of its Japanese version based on the factor loadings in the factor analysis [53]. The 13-item Japanese version of GPS was rated on a five-point Likert scale ranging from (1) *strongly disagree* to (5) *strongly agree*. A higher score refers to a higher level of procrastination tendency. Internal consistency of the GPS was adequate (α = 0.87, 95%CI = [0.86, 0.88]). The sample items of the GPS are as follows: "I often find myself performing tasks that I had intended to do days before" and "I generally delay before starting on work I have to do."

**Parental styles of caregivers.** Parental styles of caregivers were measured using the Japanese version of the Parental Bonding Instrument (PBI) [54, 55] in Sample B. The PBI contains 25 items that retrospectively measure the parental attitude of caregivers. It consists of two factors: care and over-protection. A high score on the care factor reflects supportive parenting attitudes, whereas a high score on the over-protective factor refers to unfavorable parenting attitudes. This measure asked the participants to report the attitudes of their childhood caregivers on a four-point Likert scale ranging from (0) *not true at all* to (3) *very true*. Internal consistency of each subscale was adequate (α = 0.93, 95% CI = [0.92, 0.93] for Care; α = 0.86, 95% CI = [0.84, 0.87] for Over-protection). The sample items of the PBI are as follows: "Spoke to me in a warm and friendly voice" and "Tried to control everything I did."

**Social support.** Social support was measured with the Japanese version of the Multidimensional Scale of Perceived Social Support (MSPSS) [35, 56] in Sample F. The MSPSS is a 12-item scale measuring perceived support from family, friends, and significant others. Participants answered the items on a seven-point Likert scale ranging from (1) *strongly disagree* to (7) *strongly agree*. Higher scores reflect more social support from family, friends, and significant others. Internal consistency of each subscale was adequate (α = 0.96, 95% CI = [0.95, 0.96] for family; α = 0.95, 95% CI = [0.95, 0.96] for friends; α = 0.94, 95% CI = [0.94, 0.95] for significant others). The sample items of the MSPSS are as follows: "My family really tries to help me," "I can count on my friends when things go wrong," and "There is a special person who is around when I am in need."

**Adult attachment.** Attachment style to a romantic partner was measured using the Japanese version of the Experience in Close Relationship-Relationship Structure (ECR-RS) [57, 58]

in Sample E. This scale consists of 9 items and was rated on a seven-point Likert scale ranging from (1) *strongly disagree* to (7) *strongly agree*. Six items of the nine measured attachment avoidance dimension, and the remaining three items measured the attachment anxiety dimension. Higher scores reflect higher level of attachment anxiety and avoidance, that is, more insecure attachment. Internal consistency was acceptable ($\alpha$ = 0.95, 95% CI = [0.95, 0.96] for attachment anxiety; $\alpha$ = 0.84, 95% CI = [0.83, 0.86] for the attachment avoidance). The sample items of the ECR-RS are as follows: "I don't feel comfortable opening up to this person" and "I'm afraid that this person may abandon me."

**Sociosexual orientation.**   Sociosexual orientation was measured with the Japanese version of the Revised Sociosexual Orientation Inventory (SOI-R) [59, 60] in Sample E. The SOI-R consists of 9 items measuring global sociosexual orientation, though it can also be used to measure the three subscales: sociosexual attitude, sociosexual behavior, and sociosexual desire. Because we aimed to investigate the associations between *romantic partner attachment* of the K-SF-42-J and global sociosexual orientation in this study, we only calculated the total score of unrestricted sociosexuality. This scale asked the participants to answer the SOI-R on a nine-point Likert scale. A higher score reflects higher level of unrestricted sociosexuality. Internal consistency was adequate ($\alpha$ = 0.87, 95% CI = [0.85, 0.88]). The sample items of the SOI-R are as follows: "With how many different partners have you had sex within the past 12 months," "I can imagine myself being comfortable and enjoying "casual" sex with different partners," and "In everyday life, how often do you have spontaneous fantasies about having sex with someone you have just met."

**Long-term mating orientation.**   Long-term mating orientation was measured with the Japanese version of the long-term mating orientation scale (LTMO) [41, 61] in Sample E. The LTMO contains 7 items and was rated on a seven-point Likert scale ranging from (1) *strongly disagree* to (7) *strongly agree*. A higher score reflects higher level of long-term mating orientation. Internal consistency was adequate ($\alpha$ = 0.92, 95% CI = [0.91, 0.92]). The sample items of the LTMO are as follows: "I am interested in maintaining a long-term romantic relationship with someone special" and "I would like to have a romantic relationship that lasts forever."

**Altruistic behavior.**   Altruistic behavior was measured using the Self-Report Altruism Scale Distinguished by the Recipient (SRAS-DR) [62] in Sample F. The scale consists of 21 items, including measures for altruistic behaviors toward three different targets: family members, friends/acquaintances, and strangers. Each of the three subscales includes 7 items. Participants were asked to answer the scale using a five-point Likert scale ranging from (1) *never* to (5) *very often*. A higher score represents higher frequency of each altruistic behavior. Internal consistency of each subscale was adequate ($\alpha$ = 0.90, 95% CI = [0.89, 0.90] for family members; $\alpha$ = 0.86, 95% CI = [0.85, 0.87] for friends/acquaintances; $\alpha$ = 0.90, 95% CI = [0.89, 0.91] for strangers). The sample items of the SRAS-DR are as follows: "I listen to the problems and complaints of friends or acquaintances," "I help a family member when he/she is not feeling well," and "When a stranger is looking for something, I talk to her/him."

**Spirituality.**   Spirituality was measured with the Self-Transcendence Scale (STS) [63] in Sample C. The STS contains 18 items and was rated on a seven-point Likert scale ranging from (1) *disagree* to (5) *agree*. A higher score reflects higher level of spirituality. Internal consistency was adequate ($\alpha$ = 0.94, 95% CI = [0.94, 0.95]). The sample items of the STS are as follows: "I think that we are tied with invisible bonds," "Even if I die, I think I can live as a part of nature," "I feel that I was born in this world and it has a great meaning," and "I would like to share my joy and suffering with many people."

**Demographic information.**   We asked about the participants' age (continuous) and sex (0 = Men, 1 = Women).

## Statistical analyses

First, a confirmatory factor analysis (CFA) was conducted to evaluate the seven-factor structure of the original K-SF-42. We assessed the factor structure (one-factor or seven-factor model) of the K-SF-42-J for all the participants. We evaluated the model fit of the current CFA model using the Bentler-Bonett Comparative fit index (CFI) [64], Tucker-Lewis index (TLI) [65], Root-mean-square error of approximation (RMSEA) [66], and standardized root mean-square residual (SRMR) [67]. Values of CFI and TLI greater than 0.90 are considered satisfactory levels of practical goodness-of-fit; RMSEA values of 0.06 or less are considered indications of good fit, values between 0.06 and 0.08 are considered indications of a mediocre fit, and values greater than 0.08 are considered indications of a poor fit; and SRMR values of 0.08 or less are considered indications of good fit, values between 0.08 and 0.10 are considered indications of a mediocre fit, and values greater than 0.10 are considered indications of a poor fit [68].

Since the total sample included a sample of men and women with a wide range of ages, from 20 to 59 years old, a multi-group CFA was also conducted on the K-SF-42-J, with the sample split by sex and age groups, to assess measurement invariance. The sample was split into two sex groups (men and women) and four age groups (20–29, 30–39, 40–49 and 50–59 years old), and we fitted configural, metric/weak (factor loading equality constraint), scalar/strong (factor loading and intercept equality constraint), and strict invariance models (factor loading, intercept, and residuals equality constraint) using a multi-group CFA approach [69]. Since the chi-square test is highly sensitive to sample size, we preferentially referred to both the absolute and the relative fit indices for model comparison. The absolute fit indices include CFI [64], TLI [65], RMSEA [66], and SRMR [67]. According to Chen, a decrease of less than 0.010 for CFI, an increase of less than 0.015 for RMSEA, or an increase of less than 0.030 for SRMR implies significant deterioration in model fit under testing of measurement invariance [70]. In addition, when testing scalar or strict invariance, that is at the intercept and residual variance levels, an increase of less than 0.010 is recommended for SRMR. The relative fit indices include Akaike's information criterion (AIC) [71] and Bayesian information criterion (BIC) [72]. Both the AIC and BIC provide some advantages; for example, AIC is when dealing with potential model over-fitting, as is often the case with analyses with small sample sizes [73]. Alternatively, BIC is recommended for more conservative model comparisons [74]. Since AIC and BIC have different advantages, it is generally convenient to conduct a model comparison with both fit indices. It is also worth noting that even though the model selection is based on lower AIC and BIC values, additional estimates should be computed, such as the difference (i.e., delta) between the model with the lowest AIC/BIC value and the competing models. Moreover, AIC weights and BIC weights should also be estimated. These metrics indicate the likelihood that at least one of the models under consideration is the best model.

Then we investigated the internal consistency and validity of the K-SF-42-J. The internal consistency of each K-SF-42-J subscale was checked using the Cronbach's α coefficient. Regarding the validity, we conducted the correlation analyses among related variables to show nomological network of the K-SF-42-J. All analyses were carried out in R 3.5.3, an open source statistical software environment. The *lavaan* package [75] was used for the CFA in structural equation modeling. The package *qpcR* was used for computing AIC weights and BIC weights [76]. The present data and R code were openly available at <https://osf.io/3nmxv/>.

## Results

### Confirmatory factor analysis on the K-SF-42-J and measurement invariance

Before conducting the CFA, we reversed the item scores of problems in *romantic partner attachment* in the K-SF-42-J to estimate *secure* adult romantic attachment. And we also

standardized each item score into $z$ score (*Mean* = 0, *SD* = 1) using its mean and standard deviation. After that, we tried to replicate the seven-factor structure of the original K-SF-42. In the replication of original factor structure, we compared the seven-factor (original) with the one-factor model, which is assumed in the Mini-K, to exclude the possibility that a general life history strategy factor largely accounted for the variances of item responses. The CFA of the K-SF-42-J scores indicated that the seven-factor model showed a better fit than the one-factor model ($\chi^2$ = 8293.542, *df* = 798, $p < 0.001$; CFI = 0.850; TLI = 0.838; RMSEA = 0.081; and SRMR = 0.065 for the seven-factor model; $\chi^2$ = 33085.324; *df* = 819; $p < 0.001$; CFI = 0.355; TLI = 0.322; RMSEA = 0.165; SRMR = 0.163 for the one-factor model). Though the RMSEA and the SRMR of the seven-factor model were acceptable for the cut-off criteria [68], the CFI and the TLI were not adequate. The modification indices suggested residual covariances between the unique variance of items 5 and 6 (in *Insight* factor), items 7 and 8 (in *Altruism* factor), items 11 and 12 (in *Altruism* factor), items 25 and 26 (in *Parent* factor), and items 32 and 33 (in *Family* factor). The present study modified the tested seven-factor model by adding the five residual covariances. The modified seven-factor model showed an acceptable fit (CFI = 0.912; TLI = 0.904; RMSEA = 0.062; and SRMR = 0.064). In the seven-factor model, all standardized factor loadings were over 0.4 (0.426–0.916). The detailed parameter estimates of factor loadings in the seven-factor model are shown in Table 2. All seven factors were positively and significantly correlated with each other. The detailed inter-factor correlations are provided in Table 3.

Historically, since the K-Factor was first introduced [7], it has always shown a latent hierarchical structure, with a general factor at the apex of that hierarchy [77, 78]. This is not the same thing as a "single, unidimensional K-factor," but instead what was actually reported on numerous occasions [7, 8]. We therefore performed a higher-order factor analysis based on the inter-factor correlation data among the lower-order factors presented in Table 3. As expected, we did find a general, higher-order K-Factor underlying all these lower-order factors that explained 81% of their reliable variance, and featuring the following factor loadings of the higher-order factor on these lower-order indicators. The detailed factor loadings are shown in Table 4.

Multi-group CFA was conducted on the modified seven-factor model to assess measurement invariance across sexes (Men vs. Women) and age groups (20s, 30s, 40s, vs. 50s). The detailed model fit indices are shown in Table 5. The model comparison for multi-group measurement invariance between men and women revealed that the metric invariance was supported in accordance to the minimum values of AIC and RMSEA, the maximum value of TLI, and a sizeable AIC weight. It is also worth noting that the scalar invariance was supported based on the minimum BIC value and the perfect BIC weight. Based on the cutoff values of change in model fit indices for measurement invariance test [70], the present results supported strict invariance of the K-SF-42-J across sexes (ΔCFIs < 0.010; ΔTLIs < 0.010; ΔRMSEAs < 0.010; and ΔSRMRs ≤ 0.010). The model comparison for multi-group measurement invariance across age groups revealed that the scalar invariance was supported in accordance to the minimum values of AIC and RMSEA, the maximum value of TLI, and the perfect AIC weight. Furthermore, the strict invariance was supported based on the minimum BIC value and the perfect BIC weight. Based on the cutoff values of change in model fit indices for measurement invariance test [70], the present results supported strict invariance of the K-SF-42-J across age groups (ΔCFIs < 0.010; ΔTLIs < 0.010; ΔRMSEAs < 0.010; and ΔSRMRs < 0.010).

## Descriptive statistics

Descriptive statistics for each variable were calculated. No large, biased distributions (i.e., ceiling or floor effects) were found on the variables (Table 6). We calculated the unit-weighted

**Table 2. Standardized estimates of factor loadings in CFA model.**

| | Item | Estimate | 95%CI | p |
|---|---|---|---|---|
| | *Insight, planning, and control* | | | |
| 1 | When faced with a bad situation, I do what I can to change it for the better. | 0.68 | [0.65, 0.71] | <0.001 |
| 2 | When I encounter problems, I don't give up until I solve them. | 0.81 | [0.78, 0.83] | <0.001 |
| 3 | I find I usually learn something meaningful from a difficult situation. | 0.82 | [0.80, 0.84] | <0.001 |
| 4 | When I am faced with a bad situation, it helps to find a different way of looking at things. | 0.72 | [0.69, 0.75] | <0.001 |
| 5 | Even when everything seems to be going wrong, I can usually find a bright side to the situation. | 0.76 | [0.74, 0.79] | <0.001 |
| 6 | I can find something positive even in the worst situations. | 0.74 | [0.71, 0.77] | <0.001 |
| | *General altruism* | | | |
| 7 | I spend a great deal of time per month giving informal emotional support to my blood relatives. | 0.64 | [0.60, 0.67] | <0.001 |
| 8 | I contribute a great deal to the welfare and well-being of my blood relatives in the present. | 0.68 | [0.65, 0.71] | <0.001 |
| 9 | I spend a great deal of time per month giving informal emotional support to casual acquaintances (such as neighbors or people at church). | 0.88 | [0.87, 0.90] | <0.001 |
| 10 | I contribute a great deal to the welfare and well-being of my friends these days. | 0.84 | [0.82, 0.86] | <0.001 |
| 11 | I spend a great deal of time per month doing formal volunteer work at school or other youth-related institution. | 0.57 | [0.53, 0.61] | <0.001 |
| 12 | I often contribute to any other organizations, causes, or charities (including donations made through monthly payroll deductions). | 0.59 | [0.55, 0.62] | <0.001 |
| | *Religiosity* | | | |
| 13 | I'm a very religious person. | 0.74 | [0.71, 0.76] | <0.001 |
| 14 | Religion is important in my life. | 0.89 | [0.87, 0.90] | <0.001 |
| 15 | Spirituality is important in my life. | 0.43 | [0.38, 0.47] | <0.001 |
| 16 | I closely identify with being a member of my religious group. | 0.83 | [0.82, 0.85] | <0.001 |
| 17 | I frequently attend religious or spiritual services. | 0.83 | [0.81, 0.85] | <0.001 |
| 18 | When I have decisions to make in my daily life, I often ask myself what my religious or spiritual beliefs suggest I should do. | 0.84 | [0.82, 0.86] | <0.001 |
| | *Romantic partner attachment* | | | |
| 19(R) | I worry that romantic partners won't care about me as much as I care about them. | 0.68 | [0.65, 0.71] | <0.001 |
| 20(R) | I don't feel comfortable opening up to romantic partners. | 0.73 | [0.71, 0.76] | <0.001 |
| 21(R) | I want to get close to my partner, but I keep pulling back. | 0.83 | [0.81, 0.85] | <0.001 |
| 22(R) | I often want to merge completely with romantic partners, and this sometimes scares them away. | 0.81 | [0.79, 0.83] | <0.001 |
| 23(R) | I am nervous when partners get too close to me. | 0.65 | [0.62, 0.68] | <0.001 |
| 24(R) | I find that my partner(s) don't want to get as close as I would like. | 0.82 | [0.80, 0.85] | <0.001 |
| | *Mother/father relationship quality* | | | |
| 25 | How much time and attention did your biological mother give you when you needed it? | 0.68 | [0.65, 0.71] | <0.001 |
| 26 | How much effort did your biological mother put into watching over you and making sure you had a good upbringing? | 0.64 | [0.60, 0.67] | <0.001 |
| 27 | How much did your biological mother teach you about life? | 0.70 | [0.67, 0.73] | <0.001 |
| 28 | How much love and affection did your biological father give you while you were growing up? | 0.88 | [0.87, 0.90] | <0.001 |
| 29 | How much time and attention did your biological father give you when you needed it? | 0.92 | [0.90, 0.93] | <0.001 |
| 30 | How much did your biological father teach you about life? | 0.85 | [0.83, 0.87] | <0.001 |
| | *Family social contact and support* | | | |
| 31 | How much have your relatives helped you get worries off your mind? | 0.82 | [0.80, 0.84] | <0.001 |
| 32 | How much have your relatives told you that you had done something well? | 0.80 | [0.78, 0.82] | <0.001 |
| 33 | How much have your relatives told you that they liked the way you are? | 0.74 | [0.71, 0.76] | <0.001 |
| 34 | How much have your relatives shown you affection? | 0.91 | [0.90, 0.92] | <0.001 |
| 35 | How much have your relatives listened to you when you talked about your feelings? | 0.90 | [0.89, 0.92] | <0.001 |
| 36 | How much have your relatives shown interest and concern for your well-being? | 0.88 | [0.87, 0.90] | <0.001 |
| | *Friends social contact and support* | | | |
| 37 | How much have your friends helped you get worries off your mind? | 0.90 | [0.89, 0.91] | <0.001 |
| 38 | How much have your friends told you that you had done something well? | 0.91 | [0.90, 0.92] | <0.001 |
| 39 | How much have your friends told you that they liked the way you are? | 0.89 | [0.88, 0.90] | <0.001 |

*(Continued)*

**Table 2.** (Continued)

|  | Item | Estimate | 95%CI | p |
|---|---|---|---|---|
| 40 | How much have your friends shown you affection? | 0.91 | [0.90, 0.92] | <0.001 |
| 41 | How much have your friends offered to take you somewhere? | 0.84 | [0.83, 0.86] | <0.001 |
| 42 | How much have your friends shown interest and concern for your well-being? | 0.92 | [0.91, 0.93] | <0.001 |

score of the general K-SF-42-J factor using the means of the standardized sub-scale scores. Its descriptive statistics are also shown in Table 6. Then, we calculated the internal consistency indices of the subscales of K-SF-42-J after standardizing each K-SF-42-J item score into $z$ score. All the subscales showed good internal consistency coefficients (Table 3).

## Correlation analysis between the K-SF-42-J and related measures

Correlation analyses were conducted to show the validity of the K-SF-42-J. Regarding the associations with demographic variables, *Insight, planning, and control* was positively and significantly correlated with age, and *Mother/father relationship quality*, *Family social contact and support*, and *Friends social contact and support* were inversely and significantly correlated with age. In addition, women scored significantly higher than men in *Family social contact and support*, *Friends social contact and support*, and *Romantic partner attachment*. The general K-Factor score was negatively significantly correlated with age. Its average score for women was slightly higher than that for men. The detailed associations with demographic variables are presented in Table 7. The K-SF-42-J subscale scores were weakly to moderately correlated in positive direction with the Mini-K-J score, trait EI, well-being, and cultural and social resources except for the correlations between *religiosity* and trait EI and social resources. The general K-Factor score was moderately to highly correlated with these variables with positive direction. The detailed correlation coefficients are shown in Table 8.

We subsequently examined the associations of each subscale in the K-SF-42-J with related variables. *Insight, planning, and control* in the K-SF-42-J was positively and significantly correlated with effortful control and habitual exercise and was negatively and significantly correlated with procrastination. *Mother/father relationship quality* was positively and significantly associated with care factor in parental attitudes, which reflects supportive and favorable parental style, and was negatively and significantly associated with over-protection factor in parental attitudes, which reflects negative unfavorable parenting style. *Family social contact and support* in the K-SF-42-J was positively and significantly related to social support from family

**Table 3. Inter-factor correlations between latent K-SF-42-J factors and internal consistency indices.**

|  | | Correlations | | | | | | Internal consistency |
|---|---|---|---|---|---|---|---|---|
|  | Subscale | 1 | 2 | 3 | 4 | 5 | 6 | Cronbach's α |
| 1 | Insight | | | | | | | 0.89 [0.89, 0.90] |
| 2 | Parent | 0.25 [0.20, 0.31] | | | | | | 0.91 [0.91, 0.92] |
| 3 | Family | 0.27 [0.22, 0.32] | 0.60 [0.56, 0.64] | | | | | 0.94 [0.94, 0.94] |
| 4 | Friends | 0.28 [0.22, 0.33] | 0.39 [0.34, 0.43] | 0.62 [0.58, 0.65] | | | | 0.96 [0.96, 0.96] |
| 5 | Romantic | 0.34 [0.28, 0.39] | 0.28 [0.23, 0.33] | 0.31 [0.26, 0.36] | 0.24 [0.19, 0.29] | | | 0.89 [0.88, 0.90] |
| 6 | Altruism | 0.56 [0.51, 0.60] | 0.23 [0.18, 0.29] | 0.35 [0.29, 0.40] | 0.48 [0.43, 0.52] | 0.27 [0.22, 0.32] | | 0.86 [0.85, 0.87] |
| 7 | Religious | 0.23 [0.18, 0.29] | 0.12 [0.06, 0.17] | 0.24 [0.19, 0.29] | 0.28 [0.23, 0.33] | 0.11 [0.05, 0.16] | 0.61 [0.57, 0.65] | 0.89 [0.88, 0.90] |

*Note.* 95%CIs for correlations are presented in parentheses.

**Table 4. Factor loadings of the higher-order K-Factor.**

| Subscale | Factor loading |
|---|---|
| Insight | 0.50 [0.42, 0.57] |
| Parent | 0.58 [0.51, 0.65] |
| Family | 0.74 [0.67, 0.81] |
| Friends | 0.72 [0.69, 0.76] |
| Romantic | 0.41 [0.36, 0.46] |
| Altruism | 0.65 [0.56, 0.73] |
| Religious | 0.44 [0.36, 0.53] |

*Note*. 95%CIs for correlations are presented in parentheses.

members. *Friends social contact and support* in the K-SF-42-J was also positively and significantly related to social support from friends and significant others. *Romantic partner attachment* was positively and significantly correlated with long-term mating orientation, and was negatively and significantly correlated with attachment anxiety, avoidance, and unrestricted sociosexual orientation. *General altruism* in the K-SF-42-J was positively and significantly linked to altruistic behaviors. *Religiosity* was positively and significantly related to spirituality. The general K-Factor score was positively and significantly correlated with effortful control, habitual exercise, care factor in parental attitudes, social supports from family members, friends, and significant others, altruistic behaviors, and spirituality. The general K-Factor score was negatively significantly correlated with over-protective parental attitudes and avoidant attachment style. All the detailed correlation estimates are shown in Table 9.

## Discussion

The present study provides evidence for the internal consistency and validity of the Japanese version of the K-SF-42, a measurement of human life history strategy. Based on the CFA results, seven-factor structure was acceptably fitted to the present data from the K-SF-42-J, which supports its factor validity. Inter-factor correlations were statistically significant in the positive direction. These correlations and good fit of seven-factor model to the observed data suggest that the seven life history strategy domains are valid in the Japanese population. In addition, multi-group CFA demonstrated scalar/strong measurement invariance of the K-SF-42-J over the participants' sexes and age groups, which suggests that the translation of the K-SF-42 into Japanese did not limit the measurement validity of the instrument as it generalizes well across sex and age group. Moreover, the internal consistency reliabilities of K-SF-42-J subscales ranged from 0.86 to 0.96, which are consistent with the original version [22]. These findings suggest that theK-SF-42-J is empirically validated for use in Japanese samples, enabling researchers to conduct further cross-cultural comparisons with Western national samples.

All validity measures that were used in the present study mirrored previous research on life history theory and the meta-analytic findings from Figueredo et al. [6]. All seven subscale scores and the general K-Factor score showed significant and positive correlations with the Mini-K score, trait EI, psychological well-being, and cultural and social resources in childhood, except for those of *religiosity* with trait EI and social resources. These results generally support the present hypotheses 1–1, 1–2, and 1–3. Regarding *religiosity* in the K-SF-42-J, further replication is required. Japanese people tend to freely enjoy and practice different religious customs, traditions, and rituals including celebration of Christmas day, Festival of the Dead (Obon), first shrine visit of the New Year, and commemorating the dead in Buddhist temples

**Table 5. Model comparison for multi-group measurement invariance disaggregated by sex and by age.**

| Group | Model | $\chi^2$ | df | $\Delta\chi^2$ | $\Delta df$ | p | AIC | ΔAIC | AIC Rel. L.L | AIC weights | BIC | ΔBIC | BIC Rel. L.L | BIC weights | CFI | ΔCFI | TLI | ΔTLI | RMSEA | 90%CI | ΔRMSEA | SRMR | ΔSRMR |
|---|---|---|---|---|---|---|---|---|---|---|---|---|---|---|---|---|---|---|---|---|---|---|---|
| Sex | | | | | | | | | | | | | | | | | | | | | | | |
| | Model 1 | 6301.86 | 1586 | | | | 125632.48 | 4.01 | 0.14 | 0.12 | 127235.29 | 311.40 | 0.00 | 0.00 | **0.907** | | 0.899 | | 0.064 | [0.063, 0.066] | | **0.067** | |
| | Model 2 | 6381.85 | 1628 | 79.99 | 42 | <0.001 | **125628.47** | 0.00 | 1.00 | 0.88 | 127009.84 | 85.95 | 0.00 | 0.00 | 0.906 | 0.001 | **0.901** | -0.002 | **0.064** | **[0.062, 0.065]** | 0.000 | 0.077 | 0.010 |
| | Model 3 | 6550.43 | 1663 | 168.58 | 35 | <0.001 | 125727.05 | 98.58 | 0.00 | 0.00 | **126923.89** | 0.00 | 1.00 | 1.00 | 0.903 | 0.003 | 0.900 | 0.001 | 0.064 | [0.062, 0.066] | 0.000 | 0.078 | 0.001 |
| | Model 4 | 6976.03 | 1705 | 425.60 | 42 | <0.001 | 126068.65 | 440.18 | 0.00 | 0.00 | 127044.04 | 120.15 | 0.00 | 0.00 | 0.896 | 0.007 | 0.895 | 0.005 | 0.066 | [0.064, 0.067] | 0.002 | 0.080 | 0.002 |
| Age | | | | | | | | | | | | | | | | | | | | | | | |
| | Model 1 | 8691.51 | 3172 | | | | 126078.12 | 160.09 | 0.00 | 0.00 | 129283.74 | 1891.25 | 0.00 | 0.00 | **0.892** | | 0.883 | | 0.070 | [0.068, 0.071] | | **0.071** | |
| | Model 2 | 8864.36 | 3298 | 172.84 | 126 | 0.004 | 125998.96 | 80.93 | 0.00 | 0.00 | 128540.25 | 1147.76 | 0.00 | 0.00 | 0.891 | 0.001 | 0.886 | -0.003 | 0.068 | [0.067, 0.070] | -0.002 | 0.079 | 0.008 |
| | Model 3 | 8993.42 | 3403 | 129.07 | 105 | 0.055 | **125918.03** | 0.00 | 1.00 | 1.00 | 127905.72 | 513.23 | 0.00 | 0.00 | 0.891 | 0.000 | **0.889** | -0.003 | **0.068** | **[0.066, 0.069]** | 0.000 | 0.079 | 0.000 |
| | Model 4 | 9396.52 | 3529 | 403.09 | 126 | <0.001 | 126069.12 | 151.09 | 0.00 | 0.00 | **127392.49** | 0.00 | 1.00 | 1.00 | 0.885 | 0.006 | 0.888 | 0.001 | 0.068 | [0.066, 0.070] | 0.000 | 0.080 | 0.001 |

*Notes.* Sex (n = 781 for men; n = 659 for women); Age group (n = 159 for younger people [20–29 years of age]; n = 489 for upper middle people [30–39 years of age]; n = 317 for lower middle people [40–49 years of age]; n = 475 for older people [50–59 years of age]). Model 1 = Configural invariance; Model 2 = Metric invariance; Model 3 = Scalar invariance; Model 4 = Strict invariance. AIC = Akaike's Information Criterion; BIC = Bayesian Information Criterion; CFI = comparative fit index; TLI = Tucker-Lewis index; RMSEA = root-mean-square error of approximation; SRMR = standardized root mean-square residual. The number in bold means the minimum value of AIC, BIC, RMSEA, and SRMR, or the maximum value of CFI and TLI.

**Table 6. Descriptive statistics.**

| Variable | Subscale | n | Mean | SD | Median | Theoretical range |
|---|---|---|---|---|---|---|
| K-SF-42-J | | | | | | |
| | Insight | 1440 | 0.49 | 1.02 | 0.50 | −3–3 |
| | Parent | 1440 | 1.70 | 0.78 | 1.67 | 0–3 |
| | Family | 1440 | 1.20 | 0.85 | 1.00 | 0–3 |
| | Friends | 1440 | 1.03 | 0.85 | 1.00 | 0–3 |
| | Romantic | 1440 | 0.51 | 1.14 | 0.17 | −3–3 |
| | Altruism | 1440 | −0.63 | 1.11 | −0.50 | −3–3 |
| | Religious | 1440 | −0.91 | 1.25 | −1.00 | −3–3 |
| | *K-Factor* [a] | 1440 | 0.00 | 0.64 | −0.04 | −1.79–2.45 |
| Mini-K-J | | 223 | 0.05 | 0.75 | 0.00 | −3–3 |
| Trait emotional intelligence | | 223 | 3.96 | 0.71 | 4.00 | 1–7 |
| Well-being | | 223 | 1.99 | 1.05 | 2.00 | 0–5 |
| Cultural and Social Resources | | | | | | |
| | Culture | 330 | 2.18 | 0.60 | 2.17 | 1–4 |
| | Social | 330 | 2.77 | 0.70 | 2.75 | 1–4 |
| Effortful control | | 223 | 2.59 | 0.36 | 2.57 | 1–4 |
| Habitual Exercise | | 223 | 2.88 | 1.13 | 3.00 | 1–6 |
| Procrastination | | 223 | 2.78 | 0.60 | 2.92 | 1–5 |
| Parental style | | | | | | |
| | Care | 330 | 2.79 | 0.61 | 2.75 | 1–4 |
| | Over-protection | 330 | 2.24 | 0.48 | 2.23 | 1–4 |
| Social support | | | | | | |
| | Family | 220 | 4.44 | 1.45 | 4.50 | 1–7 |
| | Friends | 220 | 4.10 | 1.47 | 4.00 | 1–7 |
| | Significant others | 220 | 4.39 | 1.40 | 4.25 | 1–7 |
| Adult attachment style | | | | | | |
| | Avoidance | 221 | 3.64 | 1.16 | 3.83 | 1–7 |
| | Anxiety | 221 | 3.48 | 1.45 | 4.00 | 1–7 |
| Sociosexuality | | 221 | 3.26 | 1.60 | 2.89 | 1–9 |
| Long-term mating orientation | | 221 | 4.59 | 1.27 | 4.43 | 1–7 |
| Altruistic behaviors | | 220 | 2.83 | 0.73 | 2.95 | 1–5 |
| Spirituality | | 223 | 2.84 | 0.77 | 2.94 | 1–5 |

*Note*. a K-Factor score was calculated as a unit-weighted factor score using the means of the standardized sub-scale scores. Its range represents the actual range of the score.

[79]. A large part of the Japanese people do not believe in any specific religion [80] and consider religion to be unimportant to them [81]. Hence, religiosity in Japanese people could be different from that in other countries.

In the seven K-SF-42-J subscales, *insight*, *planning*, and *control* showed positive correlations with effortful control and habitual exercise and negative correlation with procrastination. This supports hypotheses 2–1, 2–2, and 2–3., engagement in habitual exercise was positively and significantly related to all subscales except for *romantic partner attachment*. Engagement in habitual exercise can be considered to as allocating more bioenergetic and material resources into *somatic* effort, which are characteristics of *slow* life history strategy. The positive correlations between habitual exercise and the K-SF-42-J subscales strongly support the validity of K-SF-42-J.

**Table 7. Correlation coefficients with age and sex differences in the K-SF-42-J subscales.**

|  | Correlation with age | Men |  | Women |  | Sex differences |
|---|---|---|---|---|---|---|
| Subscale | r | Mean | SD | Mean | SD | Cohen's d |
| Insight | 0.13 [0.07, 0.18] | 0.49 | 1.01 | 0.48 | 1.03 | 0.01 [−0.09, 0.11] |
| Parent | −0.07 [−0.12, −0.02] | 1.68 | 0.76 | 1.72 | 0.80 | −0.05 [−0.15, 0.05] |
| Family | −0.15 [−0.20, −0.10] | 1.12 | 0.81 | 1.29 | 0.89 | −0.20 [−0.31, −0.10] |
| Friends | −0.16 [−0.21, −0.11] | 0.92 | 0.78 | 1.16 | 0.91 | −0.28 [−0.38, −0.17] |
| Romantic | −0.04 [−0.09, 0.01] | 0.43 | 1.10 | 0.61 | 1.17 | −0.17 [−0.27, −0.06] |
| Altruism | −0.01 [−0.06, 0.04] | −0.61 | 1.13 | −0.64 | 1.08 | 0.03 [−0.08, 0.13] |
| Religious | 0.00 [−0.06, 0.05] | −0.86 | 1.22 | −0.96 | 1.28 | 0.08 [−0.02, 0.19] |
| K-Factor | −0.07 [−0.12, −0.02] | −0.04 | 0.63 | 0.04 | 0.65 | −0.13 [−0.23, −0.02] |

*Note.* 95%CIs for correlations are presented in parentheses. *n* = 781 for men and *n* = 659 for women.

As for care and over-protection factors of parenting attitudes, *mother/father relationship quality* was positively linked to the former and negatively linked to the latter. This is consistent with the present hypothesis 2–4 and supports the validity of *mother/father relationship quality* subscale. In addition, care and over-protection factors of parental attitudes are also significantly associated with *family social contact and support* in the positive and negative direction, respectively. These associations imply that people who received warm, supportive, and favorable parental investments in their childhood tend to have more contact with and support from their family members now, and that people who received cold, harsh, and unfavorable parenting in their childhood tend to have less contact with and support from their family members now. These associations are consistent with predictions from life history theory, which further supports the K-SF-42-J's validity.

As we expected, *family social contact and support* showed a positive correlation with social support from family members and *friends social contact and support* showed positive correlation with social support from friends and significant others. These associations are consistent

**Table 8. Correlation coefficients of the K-SF-42-J with the Mini-K-J, trait emotional intelligence, well-being, and cultural and social resources in childhood.**

|  | Mini-K-J |  | Trait EI |  | Well-being |  | Cultural resource |  | Social resource |  |
|---|---|---|---|---|---|---|---|---|---|---|
| Subscale | r | 95%CI | r | 95%CI | r | 95%CI | r | 95%CI | r | 95%CI |
| Insight | 0.45 | [0.34, 0.55] | 0.57 | [0.48, 0.65] | 0.26 | [0.14, 0.38] | 0.26 | [0.15, 0.35] | 0.30 | [0.20, 0.40] |
| Parent | 0.41 | [0.30, 0.52] | 0.19 | [0.06, 0.31] | 0.36 | [0.24, 0.47] | 0.39 | [0.30, 0.48] | 0.55 | [0.47, 0.62] |
| Family | 0.45 | [0.34, 0.55] | 0.28 | [0.15, 0.39] | 0.47 | [0.36, 0.57] | 0.25 | [0.15, 0.35] | 0.42 | [0.33, 0.50] |
| Friends | 0.49 | [0.39, 0.59] | 0.16 | [0.03, 0.28] | 0.36 | [0.24, 0.47] | 0.27 | [0.17, 0.37] | 0.32 | [0.22, 0.42] |
| Romantic | 0.46 | [0.35, 0.56] | 0.32 | [0.20, 0.44] | 0.33 | [0.20, 0.44] | 0.16 | [0.05, 0.26] | 0.18 | [0.08, 0.29] |
| Altruism | 0.51 | [0.41, 0.60] | 0.36 | [0.24, 0.47] | 0.22 | [0.10, 0.34] | 0.23 | [0.13, 0.33] | 0.18 | [0.08, 0.29] |
| Religious | 0.20 | [0.07, 0.32] | 0.09 | [−0.04, 0.22] | 0.22 | [0.10, 0.34] | 0.14 | [0.03, 0.24] | 0.07 | [−0.04, 0.18] |
| K-Factor | 0.65 | [0.57, 0.72] | 0.43 | [0.32, 0.53] | 0.52 | [0.41, 0.61] | 0.38 | [0.28, 0.47] | 0.45 | [0.36, 0.54] |

**Table 9. Correlation coefficients of the K-SF-42-J with the related variables.**

|  | EC | HE | PC | PA-C | PA-O | SS-Fa | SS-Fr | SS-So | AS-Av | AS-An | SOS | LTMO | ALB | SPI |
|---|---|---|---|---|---|---|---|---|---|---|---|---|---|---|
| Insight | **0.27 [0.14, 0.39]** | **0.31 [0.19, 0.43]** | **−0.27 [−0.38, −0.14]** | 0.19 [0.09, 0.29] | −0.13 [−0.23, −0.02] | 0.31 [0.19, 0.43] | 0.37 [0.25, 0.48] | 0.39 [0.27, 0.49] | −0.33 [−0.44, −0.21] | 0.01 [−0.12, 0.15] | 0.10 [−0.03, 0.23] | 0.29 [0.16, 0.40] | 0.38 [0.26, 0.49] | 0.48 [0.38, 0.58] |
| Parent | 0.07 [−0.06, 0.20] | 0.21 [0.09, 0.34] | 0.03 [−0.11, 0.16] | **0.66 [0.60, 0.72]** | **−0.36 [−0.45, −0.26]** | 0.51 [0.41, 0.60] | 0.37 [0.25, 0.48] | 0.39 [0.27, 0.50] | −0.17 [−0.30, −0.04] | 0.00 [−0.14, 0.13] | 0.09 [−0.04, 0.22] | 0.35 [0.23, 0.46] | 0.40 [0.29, 0.51] | 0.35 [0.23, 0.46] |
| Family | 0.03 [−0.10, 0.17] | 0.20 [0.07, 0.32] | −0.04 [−0.17, 0.09] | 0.44 [0.35, 0.53] | −0.28 [−0.37, −0.17] | **0.64 [0.55, 0.71]** | 0.43 [0.31, 0.53] | 0.56 [0.46, 0.64] | −0.22 [−0.34, −0.09] | 0.05 [−0.09, 0.18] | −0.01 [−0.14, 0.13] | 0.21 [0.08, 0.33] | 0.52 [0.42, 0.61] | 0.42 [0.31, 0.53] |
| Friends | 0.07 [−0.06, 0.20] | 0.25 [0.12, 0.37] | 0.01 [−0.12, 0.14] | 0.25 [0.15, 0.35] | −0.18 [−0.29, −0.08] | 0.35 [0.22, 0.46] | **0.63 [0.54, 0.70]** | **0.46 [0.35, 0.56]** | −0.16 [−0.28, −0.03] | 0.13 [0.00, 0.26] | −0.10 [−0.23, 0.04] | 0.08 [−0.05, 0.21] | 0.55 [0.45, 0.64] | 0.42 [0.30, 0.52] |
| Romantic | 0.09 [−0.04, 0.22] | 0.09 [−0.04, 0.22] | −0.07 [−0.20, 0.06] | 0.23 [0.13, 0.33] | −0.15 [−0.25, −0.04] | 0.25 [0.12, 0.37] | 0.22 [0.09, 0.35] | 0.34 [0.22, 0.46] | **−0.43 [−0.53, −0.31]** | **−0.40 [−0.51, −0.29]** | **−0.18 [−0.31, −0.05]** | **0.26 [0.13, 0.38]** | 0.25 [0.12, 0.37] | 0.25 [0.12, 0.37] |
| Altruism | 0.18 [0.05, 0.30] | 0.51 [0.40, 0.60] | 0.09 [−0.04, 0.22] | 0.10 [−0.01, 0.21] | 0.02 [−0.09, 0.12] | 0.35 [0.23, 0.46] | 0.39 [0.28, 0.50] | 0.38 [0.26, 0.49] | −0.34 [−0.45, −0.22] | 0.09 [−0.04, 0.22] | 0.03 [−0.10, 0.16] | 0.17 [0.04, 0.30] | **0.44 [0.32, 0.54]** | 0.51 [0.41, 0.60] |
| Religious | −0.03 [−0.16, 0.10] | 0.47 [0.37, 0.57] | 0.12 [−0.01, 0.25] | −0.04 [−0.15, 0.07] | 0.10 [−0.01, 0.21] | 0.21 [0.08, 0.33] | 0.18 [0.05, 0.31] | 0.20 [0.07, 0.32] | −0.21 [−0.33, −0.08] | 0.15 [0.02, 0.27] | 0.01 [−0.12, 0.15] | 0.18 [0.04, 0.30] | 0.30 [0.18, 0.42] | **0.50 [0.40, 0.59]** |
| *K-Factor* | 0.15 [0.02, 0.27] | 0.47 [0.36, 0.56] | −0.03 [−0.16, 0.10] | 0.41 [0.32, 0.50] | −0.22 [−0.32, −0.11] | 0.57 [0.47, 0.65] | 0.56 [0.46, 0.65] | 0.59 [0.50, 0.67] | −0.41 [−0.51, −0.29] | 0.00 [−0.13, 0.13] | −0.01 [−0.14, 0.12] | 0.33 [0.21, 0.44] | 0.62 [0.53, 0.69] | 0.67 [0.59, 0.74] |

*Note*. EC = effortful control; HE = habitual exercise; PC = procrastination; PA-C = care factor in parental attitudes; PA-O = over-protection in parental attitudes; SS-Fa = social support from family members; SS-Fr = social support from friends; SS-So = social support from significant others; AS-Av = avoidance in attachment style; AS-An = anxiety in attachment style; SOS = unrestricted sociosexuality; LTMO = long-term mating orientation; ALB = altruistic behaviors; and SPI = spirituality. 95% CIs for correlations are presented in parentheses.

with the present hypothesis 2–5 and support the validity of the two subscales. Additionally, social support from family members, friends, and significant others were positively and significantly correlated with all K-SF-42-J subscales. Individuals with *slow* life history strategy are considered to have more favorable environments in which they can be helped from other people. The observed associations between social supports and the K-SF-42-J are consistent with this notion and support its validity.

Regarding *romantic partner attachment*, we observed a positive correlation with long-term mating orientation and negative correlation with attachment anxiety, avoidance, and unrestricted sociosexual orientation. These observed correlations are consistent with the hypothesis 2–6. Individuals who score high in this subscale have secure attachment style. As previous research has repeatedly shown, secure attachment is linked to the orientation toward long-term, restricted mating orientation. Thus, the present findings are consistent with those findings and support its validity.

*General altruism* showed a positive correlation with altruistic behaviors, which is consistent with the present hypothesis 2–7. In addition, altruistic behaviors are positively correlated with all K-SF-42-J subscales. As people who have *slow* life history strategy tend to develop long-term cooperative relationships, the observed positive correlations between altruistic behaviors and the K-SF-42-J subscales support the construct validity of the K-SF-42-J.

Although *religiosity* in Japanese people might be slightly different from those in other countries [79–81], *religiosity* was positively linked to spirituality. This association is consistent with

the hypothesis 2–8 and supports its validity. As with altruistic behaviors, spirituality was positively associated with all K-SF-42-J subscale scores. Previous research has shown that spirituality is positively correlated with high self-esteem, future-oriented time perspective, and less loneliness feelings [63]. High self-esteem, future-oriented time perspective, and less loneliness feelings are all characteristics of slow life history indicators [6]. Therefore, it is not surprising that spirituality associates with all K-SF-42-J subscales. Rather, these positive associations support further validation of the K-SF- 42-J.

The general K-Factor score was significantly associated with many related variables (see, Tables 7 and 8). Although the K-Factor score was not associated with procrastination, anxious attachment style, and sociosexuality, those observed correlations were generally consistent with the life history theory. This means that the K-SF-42-J is a nice, concise, and well-validated measurement of life history strategies.

Although the K-SF-42-J seems to be useful to measure *slow* life history strategy in Japanese adults, the present study also has some limitations that should be considered. First, the surveys were conducted online, and the participants were recruited in single research panel. Therefore, the present study sample cannot be considered representative of the Japanese population. However, since the proportion of Japanese people using internet is approximately 80% according to the latest Communications Usage Trend Survey in Japan [82], we believe that the present sample was not too biased. Second, the K-SF-42-J reliability was assessed only in terms of internal consistency using Cronbach's α coefficient. It is better to evaluate the test-retest reliability to assess the temporal stability of the K-SF-42-J. Finally, there are many other variables that were not used in the present study for investigating the nomological network of K-SF-42-J. As the validation of psychometric scale cannot be finished in one study we should replicate and extend the present findings of the K-SF-42-J. For example, we might investigate the associations between the K-SF-42-J and other psychometric life history instruments, such as the Life History Rating Form developed by Dunkel et al [83].

## Conclusions

The present study aimed to translate the well-developed life history strategy measurement, K-SF-42, into Japanese, and to show its internal consistency and validity in Japanese context. Obtained findings suggest that the Japanese version of K-SF-42 has good internal consistency and abundant validity of life history strategy measurement. Cross-cultural study is very important in evolutionary psychological research. The K-SF-42-J developed in the present study enable us to conduct cross-cultural comparisons in life history research, which can further extend the previous findings.

## Author Contributions

**Conceptualization:** Tetsuya Kawamoto, Satoru Kiire.

**Data curation:** Tetsuya Kawamoto.

**Formal analysis:** Tetsuya Kawamoto, Mateo Peñaherrera-Aguirre, Aurelio José Figueredo.

**Funding acquisition:** Tetsuya Kawamoto.

**Investigation:** Tetsuya Kawamoto, Satoru Kiire.

**Methodology:** Tetsuya Kawamoto.

**Project administration:** Tetsuya Kawamoto.

**Software:** Tetsuya Kawamoto, Mateo Peñaherrera-Aguirre, Aurelio José Figueredo.

**Supervision:** Aurelio José Figueredo.

**Validation:** Tetsuya Kawamoto.

**Visualization:** Tetsuya Kawamoto.

**Writing – original draft:** Tetsuya Kawamoto.

**Writing – review & editing:** Satoru Kiire, Rachel Zambrano, Mateo Peñaherrera-Aguirre, Aurelio José Figueredo.

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
