## [Decision Letter · Decision Letter 0]

9 Jun 2022

PONE-D-22-10116Development and validation of a Japanese translation of the K-SF-42PLOS ONE

Dear Dr. Kawamoto,

Thank you for submitting your manuscript to PLOS ONE. After careful consideration, we feel that it has merit but does not fully meet PLOS ONE’s publication criteria as it currently stands. Therefore, we invite you to submit a revised version of the manuscript that addresses the points raised during the review process.

We look forward to receiving your revised manuscript.

Kind regards,

Sónia Brito-Costa, Ph.D.

Academic Editor

PLOS ONE

Journal Requirements:

"This research was supported by JSPS KAKENHI Grant Number JP17K13921."

Reviewers' comments:

Reviewer's Responses to Questions

**Comments to the Author**

1. Is the manuscript technically sound, and do the data support the conclusions?

Reviewer #1: Yes

Reviewer #2: Yes

2. Has the statistical analysis been performed appropriately and rigorously? 

Reviewer #1: Yes

Reviewer #2: Yes

3. Have the authors made all data underlying the findings in their manuscript fully available?

Reviewer #1: Yes

Reviewer #2: Yes

4. Is the manuscript presented in an intelligible fashion and written in standard English?

Reviewer #1: Yes

Reviewer #2: Yes

5. Review Comments to the Author

Reviewer #1: The objective of this study was to demonstrate the internal consistency and construct validity of Japanese translation of the K-SF-42 (J-K-SF-42).

This study has been well conducted. I only have a few comments and suggestions, especially with a view to clearer reporting.

1. There are several general classes of reliability estimates. In this study, the authors determined the internal consistency of the J-K-SF-42. I suggest to mention always ‘internal consistency’ instead of ‘reliability’ (see for example lines 20, 449, 495).

2. Construct validity should be assessed by testing predefined hypotheses. The authors of this paper did this thoroughly. Notwithstanding this, I challenge them to think about how they can display the confirmation of these hypotheses in a more comprehensive way, for example using a table.

3. Furthermore, I think it would be useful, when evaluating these hypotheses, not only to look at whether there is a correlation between the subscale in question and a particular scale, but also whether this correlation is stronger than the correlation of the other subscales with that scale.

4. Line 237: the participants received a reward for participation. Is it possible to give some more information on this?

5. Table 1: Since a table must be readable independently, is it possible to list all abbreviations at the bottom of the table? For example K-SF-42-J and Mini-K-J are missing.

6. Lines 267-268: is the K-SF-42-J a six- or seven-point Likert scale?

7. Lines 444-448: maybe some guidelines can be given about how the fit-indices should be interpreted?

8. Lines 465-469: Why is the AIC and BIC not used to compare the one-factor model with the seven-factor model?

9. Lines 483-487: “According to Chen, … is recommended for SRMR”: this could be moved to the method section.

10. Title table 7 and table 8: it must be possible to read the table without reading the text. For example, I suggest to change the title of table 7 into: “Correlation coefficients of the K-SF-42-J with the Mini-K-J, trait emotional intelligence, well-being, and cultural and social resources in childhood”

11.Lines 560-562: “In addition, multi-group CFA demonstrated scalar/strong measurement invariance of the K-SF-42-J over the participants’ sexes and age groups.” Explain the implications of this result.

Reviewer #2: This manuscript describes a Japanese translation of the K-SF-42, a short form of the Arizona Life History Battery (ALHB). Analyses of data from 1440 Japanese online respondents established the instrument's scale reliability and convergent validity.

As a purely methodological project, submitted to an online journal, this work should be published with minor revisions, including improving its English, something that I'm sure one of the native-English-speaking co-authors could do. (For example, there are quite a few missing grammatical articles).

There've been serious substantive challenges in the recent literature to the ALHB and its offshoots, which I'm sure the authors are aware of, but the review process for this paper isn't the place to do battle over them.

Comments and suggestions:

A seven-factor structure was a better fit to the data than a one-factor structure. What does this result say about the value of the concept of a single, unidimensional K-factor, claimed to represent human psychometric life history strategy?

Some of the instruments chosen to demonstrate convergent validity with K-SF-42-J scales seem to be *too* convergent, so to speak -- so similar to their matched K-SF-42-J scales in item content that high correlations are inevitable. This concern applies to the "Parental Styles of Caregivers," "Social Support," "Altruistic Behavior," and especially the "Adult Attachment" instruments. In view of this, some of the bold-faced correlations in Table 8, although significant, are surprisingly low.

Japan has an extraordinarily high frequency of single young adults, many of whom claim to be uninterested in intimate relationships. See for example, https://www.u-tokyo.ac.jp/focus/en/press/z0508_00142.html. This raises the problem of interpreting participants' answers to questions about adult attachment, SOI, and LTMO. What does it mean to describe your romantic attachment style if you've never had a romantic relationship, and don't want one?

There's apparently a copy-and-paste error on page 16, as the item "Seeing other people exercise motivates me to be more active" appears as an example of both the "Effortful Control" scale and the "Habitual Exercise" scale (where it presumably belongs).

The authors note that additional work should be done to validate this instrument. I would suggest creating Japanese translations of other psychometric LH instruments, such as Dunkel's Life History Rating Form, and checking whether the correlations between these instruments in Japanese is similar to their correlations in English.

6. PLOS authors have the option to publish the peer review history of their article (what does this mean?). If published, this will include your full peer review and any attached files.

Reviewer #1: **Yes: **Wim Peersman

Reviewer #2: No

---

## [Author Response · Author response to Decision Letter 0]

17 Aug 2022

Dear Professor Brito-Costa,

We wish to thank you for allowing us to revise and resubmit our manuscript (“Development and validation of a Japanese translation of the K-SF-42” [PONE-D-22-10116]). We were pleased to find that both reviewers acknowledged the relevance and potential contribution of our studies. Both reviewers also mentioned several points that they would like to see addressed in a revision. Below, we describe in detail how we have processed the reviewer’s points. By doing so the manuscript has been approved and we hope you may now find it suitable for publication in PLOS ONE.

We are looking forward to hearing from you.

Kind regards,

The authors

RESPONSE TO EDITOR:

We strongly thank the Editor for acknowledging the value of our manuscript. We received three comments from the Editor. All of our replies to them are shown in the response letter. We would appreciate it if the Editor could refer to it. We thank you again for the editor's useful comments. We trust that the revised manuscript is suitable for publication.

RESPONSE TO REVIEWER 1:

We wish to express our appreciation to the Reviewer 1 for his or her insightful comments, which have helped us to significantly improve the paper. All of our replies to them are presented in the response letter in detail. We would appreciate it if the Reviewer 1 could refer to it. We wish to thank the Reviewer 1 again for his or her valuable comments.

RESPONSE TO REVIEWER 2:

We strongly thank the Reviewer 2 for his or her fruitful suggestions. We feel the comments have helped us significantly improve the paper. All of our replies to them are shown in the response letter in detail. We would appreciate it if the Reviewer 2 could refer to it. We wish to thank the Reviewer 1 again for his or her valuable comments. Thank you again for your comments on our paper.

---

## [Editor Report · Decision Letter 1]

24 Aug 2022

Development and validation of a Japanese translation of the K-SF-42

PONE-D-22-10116R1

Dear Dr. Kawamoto,

We’re pleased to inform you that your manuscript has been judged scientifically suitable for publication and will be formally accepted for publication once it meets all outstanding technical requirements.

Kind regards,

Sónia Brito-Costa, Ph.D.

Academic Editor

PLOS ONE
---

## [Editor Report · Acceptance letter]

7 Sep 2022

PONE-D-22-10116R1 

Development and validation of a Japanese translation of the K-SF-42 

Dear Dr. Kawamoto:

I'm pleased to inform you that your manuscript has been deemed suitable for publication in PLOS ONE. Congratulations! Your manuscript is now with our production department. 

Kind regards, 

on behalf of

Dr. Sónia Brito-Costa 

Academic Editor

PLOS ONE